# Multi-Label Learning with Block Diagonal Labels

**Leqi Shen**
Tsinghua University
School of Software, BNRist
Beijing, China
lunarshen@gmail.com

**Sicheng Zhao***
Tsinghua University
BNRist
Beijing, China
schzhao@gmail.com

**Yifeng Zhang**
jd.com
Beijing, China
zhangyifeng3@jd.com

**Hui Chen**
Tsinghua University
BNRist
Beijing, China
jichenhui2012@gmail.com

**Jundong Zhou**
Tsinghua University
School of Software, BNRist
Beijing, China
jundong.zhou@outlook.com

**Pengzhang Liu**
jd.com
Beijing, China
liupengzhang@jd.com

**Yongjun Bao**
jd.com
Beijing, China
baoyongjun@jd.com

**Guiguang Ding***
Tsinghua University
School of Software, BNRist
Beijing, China
dinggg@tsinghua.edu.cn

## Abstract

Collecting large-scale multi-label data with *full labels* is difficult for real-world scenarios. Many existing studies have tried to address the issue of missing labels caused by annotation but ignored the difficulties encountered during the annotation process. We find that the high annotation workload can be attributed to two reasons: (1) Annotators are required to identify labels on widely varying visual concepts. (2) Exhaustively annotating the entire dataset with all the labels becomes notably difficult and time-consuming. In this paper, we propose a new setting, *i.e. block diagonal labels*, to reduce the workload on both sides. The numerous categories can be divided into different subsets based on semantics and relevance. Each annotator can only focus on its own subset of labels so that only a small set of highly relevant labels are required to be annotated per image. To deal with the issue of such *missing labels*, we introduce a simple yet effective method that does not require any prior knowledge of the dataset. In practice, we propose an **A**daptive **P**seudo-**L**abeling method to predict the unknown labels with less noise. Formal analysis is conducted to evaluate the superiority of our setting. Extensive experiments are conducted to verify the effectiveness of our method on multiple widely used benchmarks.

## CCS Concepts

• **Computing methodologies** → **Object recognition**.

*Corresponding author

## Keywords

Multi-label classification, Missing labels, Pseudo labels

**ACM Reference Format:**
Leqi Shen, Sicheng Zhao, Yifeng Zhang, Hui Chen, Jundong Zhou, Pengzhang Liu, Yongjun Bao, and Guiguang Ding. 2024. Multi-Label Learning with Block Diagonal Labels. In *Proceedings of the 32nd ACM International Conference on Multimedia (MM '24), October 28-November 1, 2024, Melbourne, VIC, Australia.* ACM, New York, NY, USA, 11 pages. https://doi.org/10.1145/3664647.3680793

## 1 Introduction

Image classification [21, 32] is a fundamental and important research area in the computer vision community. Unlike single-label classification, images for multi-label classification usually contain complex scenes with several objects [42–44]. Due to the extensive number of categories, collecting a realistic large-scale dataset with complete labels is notably laborious [10, 20]. Consequently, annotators inadvertently overlook certain objects [37], resulting in missing labels.

Previous works study multi-label classification with *missing labels* in the settings of *partial labels* [13] and *single positive labels* [7]. The simulation of *partial labels* annotates each image with a fixed number of categories to study unannotated labels. In order to explore the minimal supervision, only one positive label is annotated per image in *single positive labels*. Both settings are employed to investigate the unannotated labels caused by annotation while neglecting the annotation challenges encountered during the annotation process.

As shown in Fig 2.(a), the general annotation pipeline for multi-label classification is that the annotators first learn the visual concepts and differences of all categories, and then annotate a large number of images with complete labels. The high annotation workload for multi-label classification can be attributed to two reasons. (1) Learning: Annotators are required to identify labels on widely varying visual concepts, which can take a lot of effort. The large

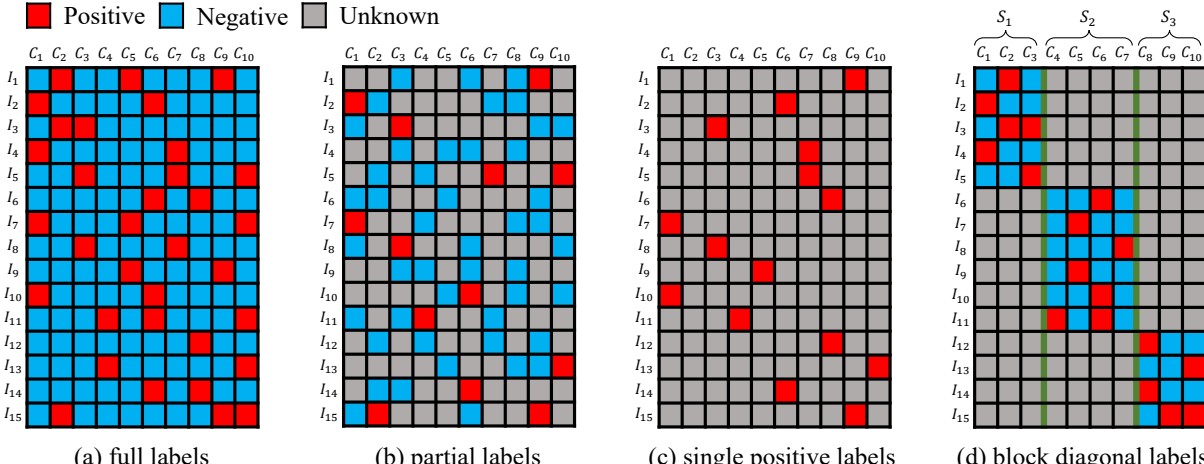

(a) full labels      (b) partial labels      (c) single positive labels      (d) block diagonal labels

**Figure 1: Illustration of *full labels, partial labels, single positive labels*, and the proposed *block diagonal labels* on a dataset with 15 images $\{I_i\}_{i=1}^{15}$ and 10 categories $\{C_j\}_{j=1}^{10}$. Each row shows the label vector of an image. The annotation matrix consists of all label vectors. For each image, all categories are annotated in (a) *full labels*, a fixed portion of random categories are annotated in (b) *partial labels*, one category is annotated with positive label in (c) *single positive labels*, and a subset of interrelated categories are annotated in (d) our proposed *block diagonal labels*.**

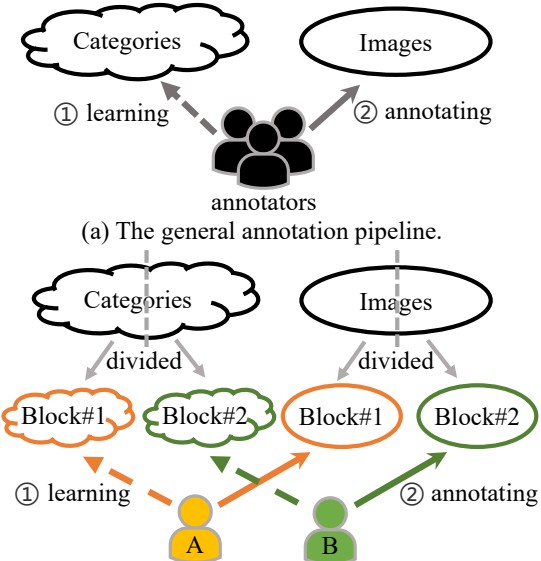

(a) The general annotation pipeline.

(b) The annotation pipeline of block diagonal labels.

**Figure 2: (a) The general annotation pipeline. The annotators first learn the visual concepts and differences of all categories and then annotate all the images. (b) The annotation pipeline of *block diagonal labels*. Categories can be partitioned into different subsets according to semantics and relevance. The annotators simply focus on their own block, so that the annotators can individually annotate a smaller dataset with a subset of highly relevant labels.**

size of the data requires a lot of highly skilled annotators. (2) Annotating: As the number of to-be-annotated images and categories grows, exhaustively annotating the entire dataset with all the labels becomes notably difficult and time-consuming.

In light of the aforementioned two aspects, we propose a new setting of multi-label classification with *block diagonal labels*. As depicted in Fig 1.(d), the *block diagonal labels* setting exhibits similarity with block diagonal matrices whose diagonal contains blocks of smaller matrices. Each block corresponds to a single coarse label. In both online resources and existing single-label datasets, numerous images are accompanied by coarse labels. Furthermore, well-trained single-label classifiers have the capacity to obtain coarse labels for unannotated images. For these reasons, coarse labels are readily available. Therefore, we propose to annotate the multi-label dataset by utilizing the prior coarse label present in each image. Fig 3 shows some image examples with coarse labels, where categories in the block are labeled.

The annotation process for *block diagonal labels*, as illustrated in Fig 2.(b), consists of two stages: (1) All categories are partitioned into distinct coarse labels according to their semantic similarities. The unannotated images are grouped based on their respective coarse label. These categories and images, which belong to the same coarse label, form a specific block in Fig 1.(d). Each annotator is responsible for only one of these blocks, learning the semantic knowledge of the labels within that block. (2) Annotators individually perform annotations of their designated block. For instance, the annotator *A* can only focus on Block#1 in Fig 2.(b).

The primary advantage of *block diagonal labels* is that annotators can purely concentrate on their assigned block, which contains a compact subset of highly relevant labels, significantly smaller than the complete set of labels. As a result, training annotators to learn visual concepts is more efficient, and the number of annotated labels required per image is greatly reduced. Therefore, the proposed *block diagonal labels* setting can readily obtain a large-scale dataset with a great number of labels.

Unknown labels are still the core problem in the *block diagonal labels* setting. We propose a simple yet effective method to deal with the problem of unknown labels. A naive approach is to use

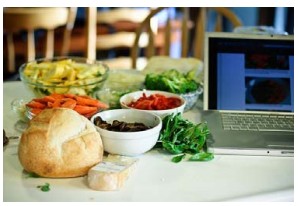 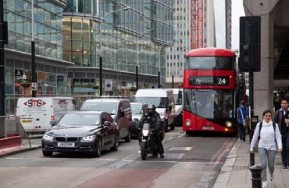 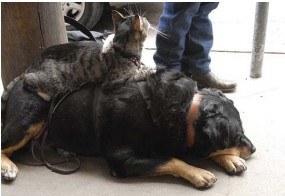 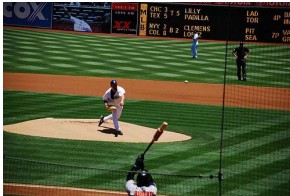

Coarse Label: food
Block Diagonal Labels :
    banana, broccoli, carrot

Full Labels:
    bowl, banana, broccoli, carrot,
    chair, dining table, laptop

Coarse Label: vehicle
Block Diagonal Labels :
    car, motorcycle, bus, truck

Full Labels:
    person, backpack, handbag, car,
    motorcycle, bus, truck, traffic light,
    fire hydrant

Coarse Label: animal
Block Diagonal Labels :
    cat, dog

Full Labels:
    person, car, cat, dog

Coarse Label: sports equipment
Block Diagonal Labels :
    sports ball, baseball bat, baseball glove

Full Labels:
    person, sports ball, baseball bat,
    baseball glove, tv

**Figure 3: Example of images in COCO2014 with the *full labels* and *block diagonal labels* settings. In *full labels,* all categories are labeled in each image. As for *block diagonal labels,* images have a coarse label prior. The categories belonging to the same coarse label as the image are labeled.**

a hard threshold to distinguish true negative labels, but it will inevitably introduce some noise. Thus, we propose an adaptive pseudo-labeling method that dynamically adjusts the threshold to reduce the noise for labeling the unknown labels. Any relationships between categories or prior statistics of data are not involved in our method, which can be regarded as a strong baseline.

To evaluate the effectiveness of our proposed setting, we conduct a formal analysis of the annotation workload. In addition, we compare with state-of-the-art methods in our setting. We artificially create the *block diagonal labels* datasets from VOC2012 [14], COCO2014 [23], NUSWIDE [6]. We extend this evaluation on a more extensive scale, utilizing the large-scale dataset OpenImages [22]. The block divisions are modified according to the category hierarchies of the published papers. Our method significantly outperforms the existing baselines and achieves state-of-the-art performance: 81.92% mAP on COCO2014 with 9.31% annotated labels and 63.26% mAP on NUSWIDE with 15.60% annotated labels.

The contributions are summarized in three-fold: (1) We argue the difficulty of producing realistic large-scale multi-label datasets from two aspects: learning and annotating, which are neglected in *partial labels* and *single positive labels*. (2) We propose a new multi-label classification setting with *block diagonal labels* where annotators can simply concentrate on their assigned block to reduce the annotation workload and positive labels are guaranteed per image. (3) We propose a strong baseline to adaptively predict unknown labels without any prior of datasets or categories, which achieves state-of-the-art performance.

## 2 Related Work

### 2.1 Multi-label Learning with Full Labels

There are a lot of remarkable works in the field of multi-label learning. An important direction is to model correlation between labels via graph neural networks [2, 4, 5, 39] or transformer structure [25, 31]. [31] predicts labels through queries and extracts local discriminative features adaptively for different labels. Another key

characteristic of this field is the inherent positive-negative imbalance. [29] dynamically focuses on the hard samples and controls the contribution propagated from the positive and negative samples.

### 2.2 Multi-label Learning with Missing Labels

The setting of [40] significantly differs from ours. In [40], the candidate set (positive labels) for each image includes both relevant (true positive) and irrelevant (noisy) labels. In contrast, more recent studies have focused on *missing labels*, where not all positive labels are annotated. Many works [13, 17, 46] focus on the optimization of the loss function. [13] is the first work to train a deep neural network for *partial labels*, which introduces partial-binary cross entropy. [17] proposes to reject or correct the large loss samples to prevent the model from memorizing the noisy label. [46] proposes Hill to down-weight negatives and self-paced loss correction to correct potential unknown labels. There are some works [1, 7, 16] that estimate unknown labels by the label prior or correlation. Some other works [3, 13, 27, 34] require extra complex architectures.

Different from the exiting *missing labels* setting, the novel *block diagonal labels* setting is introduced to take the annotators' learning and annotating process into consideration. Instead of modeling label correlation and using the dataset priors or extra architectures, we propose to adaptively predict unknown labels by dynamically adjusting the thresholds in the training process.

## 3 Block Diagonal Labels

### 3.1 Comparison with Other Settings

In order to reduce the annotation workload, recent works mainly focus on the *missing labels* setting, which annotates a small subset of positive and negative labels. We demonstrate the difference between the *full labels* and existing *missing labels* settings based on the example in Fig 1. The dataset has 15 images $\{I_i\}_{i=1}^{15}$ and 10 categories $\{C_i\}_{i=1}^{10}$. In the *full labels* setting, all categories are completely annotated for a given image. Positive labels are marked in red and negative labels are marked in blue. Each row shows the label vector for each image. Fig 1.(a) illustrates the annotation matrix with the *full labels* setting.

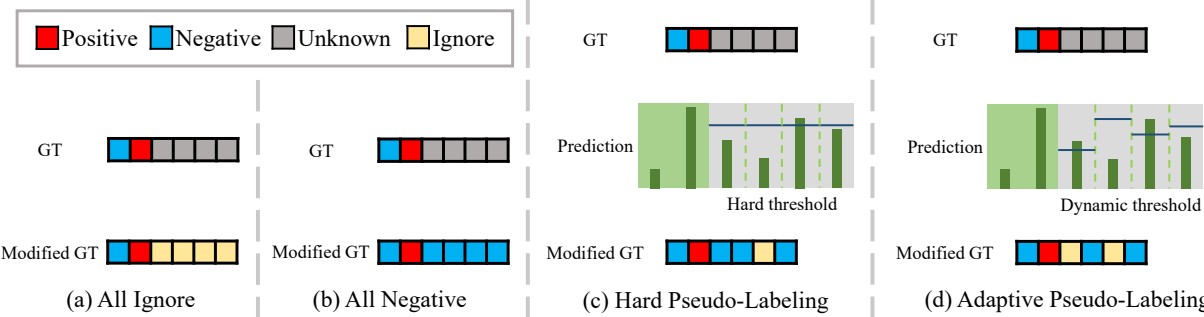

**Figure 4: Different strategies for handling unknown labels given an example with ground-truth labels. (a) AI ignores all unknown labels whose loss will not be calculated. (b) AN assumes all unknown labels as negative. (c) HPL utilizes hard thresholds to find the true negative labels. (d) APL applies dynamic thresholds to obtain negatives with less noise.**

As for the *missing labels* setting, the unknown labels are annotated in gray. The *partial labels* and *single positive labels* settings for *missing labels* are widely studied by previous works. Fig 1.(b) shows the *partial labels* setting, which annotates each image with a small number of labels. Related works manually generate the *partial labels* dataset, where each image is randomly selected with a fixed portion of categories as annotations and the others are assumed to be unknown. Since the number of positive labels per image is small on the existing dataset, *partial labels* will inevitably produce images without positive labels, which does not match the true *missing labels* setting. Furthermore, Fig 1.(c) shows the *single positive labels* setting, which annotates one positive label per image, and no additional negative or positive labels are provided. This setting has the least supervision on the multi-label task. However, the scarce supervision greatly limits the performance.

Our proposed *block diagonal labels* setting also belongs to *missing labels*. In each block, the images have the same coarse label as the category subset. In Fig 1.(d), each image is annotated with only one subset of categories. The labels in the same subset are correlated and the number of labels in each subset is small. This facilitates annotators in easily annotating the specified subset of labels. Accordingly, we assume in the experiment that labels within each subset are fully annotated. When subset $S_i$ is annotated by its annotator, the other subsets $\{S_j | j \neq i, j = 1, 2, 3\}$ are ignored. As a result, labels of other subsets are considered unknown.

In practical applications, an image may contain multiple coarse labels, each of which can be simultaneously annotated by different annotators. Each annotator is responsible for their designated block, enabling the parallel processing of annotations. These findings suggest that our setting is suitable for complex real-world label distributions. However, our study on *block diagonal labels* investigates a more extreme scenario, where an image is associated with only a single coarse label, as illustrated in Fig 1.(d).

## 3.2 Analysis on Annotation Workload

To demonstrate the superiority of our *block diagonal labels*, we conduct simulation and analysis of different settings, given a dataset of $N$ images and $M$ categories which are divided into $O$ blocks. $N_i$ indicates the number of images in block $i$ and $M_i$ indicates the number of categories in block $i$. Each block requires at least one

annotator, so we make the assumption that there are a total of $O$ annotators.

From learning, we assume that the cost of learning category $i$ is $p_i$. In the other settings, all annotators are required to learn all the categories, resulting in a cost of $O \times \sum_{i=1}^{M} p_i$. However, in *block diagonal labels*, each annotator only learns the categories within their assigned block, significantly reducing the cost to $\sum_{j=1}^{O} \sum_{i=1}^{M_j} p_i$. The comparison is as follows:

$$\sum_{j=1}^{O} \sum_{i=1}^{M_j} p_i < O \times \max \left\{ \sum_{i=1}^{M_j} p_i : j = 1, \cdots, O \right\}$$
$$< O \times \sum_{i=1}^{M} p_i. \tag{1}$$

From annotating, we assume that the cost of annotating category $i$ is $q_i$. In *full labels*, the cost is $N \times \sum_{i=1}^{M} q_i$. In *block diagonal labels*, the cost is $\sum_{k=1}^{O} \sum_{j=1}^{N_k} \sum_{i=1}^{M_k} q_i$. The comparison is as follows:

$$\sum_{k=1}^{O} \sum_{j=1}^{N_k} \sum_{i=1}^{M_k} q_i < \sum_{k=1}^{O} \sum_{j=1}^{N_k} \max \left\{ \sum_{i=1}^{M_l} q_i : l = 1, \cdots, O \right\}$$
$$= N \times \max \left\{ \sum_{i=1}^{M_l} q_i : l = 1, \cdots, O \right\} < N \times \sum_{i=1}^{M} q_i. \tag{2}$$

In the Appendix, we conduct additional experiments for *partial labels* and *block diagonal labels* under conditions with a similar number of annotated labels.

Exiting *partial labels* and *single positive labels* settings only consider missing labels resulting from annotation errors, but ignore the cost during the annotation pipeline. Our proposed setting of *block diagonal labels* reduces the annotation workload from both learning and annotating.

## 4 Methodology

### 4.1 Problem Formulation

A dataset consists of images $\{I_i | i = 1, 2, \cdots, N\}$ and ground truth labels $\{y_i | i = 1, 2, \cdots, N\}$, where $N$ is the number of training samples. $y = \{c_i | i = 1, 2, \cdots, M\}$, and $M$ is the number of categories. In *full labels*, the category $c_i$ is labelled by positive '1' or negative

'-1'. As for *missing labels*, $c_i \in \{1, -1, 0\}$, where 0 is an "unknown" annotation.

In our *block diagonal labels* setting, all categories are divided into subsets $\{S_i | i = 1, 2, \cdots, k\}$ based on the semantic and relevance, where $k$ is the number of subsets. $s_i = |S_i|$, $\sum_{i=1}^{k} s_i = M$, where $s_i$ is the size of subset $S_i$. The annotators only annotate $S_i$ of their own block $B_i$. The final dataset $D$ is the union of all block datasets $\{D_{B_i} | i = 1, 2, \cdots, k\}$. The label vector of an image in block $B_j$ is as follows:

$$c_i = \begin{cases} 1 \ or -1, & c_i \in S_j, \\ 0, & c_i \notin S_j. \end{cases} \tag{3}$$

The annotation number per image in block $B_j$ is $s_j$, where $s_j \ll M$. Unknown labels are still the primary concern requiring resolution in *block diagonal labels*.

Given an image $I$ with *missing labels*, its label vector $y$ is composed of positive labels $P_I = \{i | c_i = 1\}$, negative labels $N_I = \{i | c_i = -1\}$, and unknown labels $U_I = \{i | c_i = 0\}$, where $|P_I| + |N_I| + |U_I| = M$. There are two naive baselines that can be applied in *missing labels*, which show different strategies for the unknown labels.

**All Ignore (AI).** In this baseline, we simply ignore the unknown labels. In Fig 4.(a), only positive and negative labels are used as supervisory signals to calculate the loss:

$$L(I) = \sum_{c \in P_I} L^+(p_c) + \sum_{c \in N_I} L^-(p_c), \tag{4}$$

where the prediction output for category $c$ is $p_c$. The ignored labels do not compute gradients to optimize the network and the predictions of those are seen as the ground truth, which misses valuable training signals.

**All Negative (AN).** In a typical multi-label dataset, there are far more negative labels than positive labels. Therefore, we assume all the unknown labels as negative labels in Fig 4.(b). The loss function is as follows:

$$L(I) = \sum_{c \in P_I} L^+(p_c) + \sum_{c \in N_I \cup U_I} L^-(p_c). \tag{5}$$

There are also some false negatives of the unknown labels, which leads the model to predict the true positive labels incorrectly.

**Loss Function.** The loss function in AI and AN is the Binary Cross Entropy loss, which is defined as:

$$\begin{cases} L_{BCE}^+(p_c) = -log(p_c), \\ L_{BCE}^-(p_c) = -log(1 - p_c). \end{cases} \tag{6}$$

However, the severe imbalance between positive labels and negative labels is an important problem for multi-label learning [38]. PASL loss [1] is a modified ASL loss [29] to mitigate the imbalance problem for *missing labels*:

$$\begin{cases} L_{PASL}^+(p_c, \gamma) = -(1 - p_c)^\gamma log(p_c), \\ L_{PASL}^-(p_c, \gamma) = -(p_c)^\gamma log(1 - p_c), \end{cases} \tag{7}$$

where $\gamma$ is a focusing parameter, which performs more attention to the hard samples with a larger value. In the following paper, PASL loss is the base loss function whose abbreviation is $\{L^+(p_c, \gamma), L^-(p_c, \gamma)\}$.

PASL is based on AN:

$$\begin{aligned} L(I) = \sum_{c \in P_I} L^+(p_c, \gamma^+) + \sum_{c \in N_I} L^-(p_c, \gamma^-) \\ + \sum_{c \in U_I} L^-(p_c, \gamma^u), \end{aligned} \tag{8}$$

where $\gamma^+ < \gamma^- < \gamma^u$. Unknown labels are the most frequent and have less confidence than negative labels.

## 4.2 Hard Pseudo-Labeling (HPL)

A naive approach to reduce the false negatives is to use a constant threshold for all categories. If the prediction probability for an unknown label is below the pre-defined threshold, it is likely a true negative. HPL computes a pseudo label for each unknown label using the predictions:

$$\begin{aligned} U_I^{Neg} = \{c_i | c_i \in U_I, p_{c_i} <= \tau\}, \\ U_I^{Ign} = \{c_i | c_i \in U_I, p_{c_i} > \tau\}, \end{aligned} \tag{9}$$

where $U_I = U_I^{Neg} \cup U_I^{Ign}$, and $\tau$ is a pre-defined hard threshold. As illustrated in Fig 4.(c), the unknown labels are partitioned by a hard threshold. Then, the overall loss function is calculated as follows:

$$\begin{aligned} L(I) = \sum_{c \in P_I} L^+(p_c, \gamma^+) + \sum_{c \in N_I} L^-(p_c, \gamma^-) \\ + \sum_{c \in U_I^{Neg}} L^-(p_c, \gamma^u), \end{aligned} \tag{10}$$

where $U_I$ is replaced by $U_I^{Neg}$ in Eqn. (8). Like [1], $U_I^{Ign}$ is not involved in the final loss, where the positive and negative are really hard to distinguish.

## 4.3 Adaptive Pseudo-Labeling (APL)

[45] indicates that using identical hard thresholds for all categories fails to consider different learning statuses and learning difficulties of different categories. Thresholds should be flexibly updated for each category during the training stage. Unlike [45] for single-label classification, we maintain memory banks of positive predictions to compute the learning status of categories: $MB = \{mb_1, mb_2, \cdots, mb_M\}$, where $mb_i$ is the memory bank of category $c_i$. We store the positive predictions of the past $Z$ mini-batch, where $mb_i = \{p_{c_i}^{avg_z}\}_{z=1}^{Z}$ is updated with first-in-first-out strategy. $p_{c_i}^{avg_z}$ is the average predictions of positive $c_i$ in $z$ mini-batch. Note that we only count the predictions of true positive labels $P$. Then, the learning status of $c_i$ is formulated as follows:

$$\alpha_{c_i} = \frac{1}{|mb_i|} \sum_{p \in mb_i} p, \tag{11}$$

where $\alpha_{c_i} \in [0, 1]$. Since negative samples are more frequent compared to positive samples, the model tends to predict them with low probabilities. Larger $\alpha_{c_i}$ indicates that the model can better classify the category $c_i$. The flexible threshold is defined as:

$$\tau_{c_i} = \tau_{bottom} + (\tau_{top} - \tau_{bottom}) \cdot \alpha_{c_i}, \tag{12}$$

**Table 1: Comparison with state-of-the-arts in *block diagonal labels*. The best and second-best results are marked in bold and underlined, respectively. EMA indicates the exponential moving average strategy.**

| Backbone | Method | VOC2012 | | | COCO2014 | | | NUSWIDE | | |
|---|---|---|---|---|---|---|---|---|---|---|
| | | mAP | CF1 | OF1 | mAP | CF1 | OF1 | mAP | CF1 | OF1 |
| TResNet-Large | AI | 82.01 | 80.71 | 77.94 | 67.30 | 66.76 | 61.75 | 47.70 | 50.77 | 54.61 |
| | AN | 84.48 | 80.04 | 81.69 | 70.54 | 67.31 | 69.69 | 55.96 | 57.72 | 70.43 |
| | ASL [29] | 90.29 | 86.19 | 87.33 | 75.04 | 71.32 | 73.66 | 58.02 | 59.42 | 70.69 |
| | SPLC [46] | 90.42 | 87.07 | 88.58 | 75.90 | 72.45 | 74.70 | 58.31 | 59.52 | 71.96 |
| | LL [17] | 87.28 | 82.84 | 85.43 | 75.81 | 72.22 | 72.49 | 57.78 | 58.69 | 71.71 |
| | PASL [1] | 90.47 | 86.57 | 87.87 | 76.39 | 72.51 | 74.78 | 58.87 | 59.88 | 71.65 |
| | Selective [1] | 87.31 | 84.68 | 80.68 | **78.32** | **75.00** | 76.57 | 57.77 | 59.04 | **72.18** |
| | MLdecoder [31] | 89.39 | 85.67 | 86.70 | 76.02 | 72.07 | 74.63 | 57.90 | 59.63 | 71.20 |
| | BoostLU [18] | 87.71 | 83.89 | 85.89 | 76.09 | 72.77 | 73.05 | 57.71 | 59.74 | 71.87 |
| | APL (Ours) | **91.56** | **88.21** | **89.36** | 78.25 | 74.61 | **76.73** | **59.69** | **60.35** | 72.12 |
| | APL-EMA (Ours) | 91.56 | 88.21 | 89.36 | 81.92 | 77.99 | 80.05 | 63.26 | 62.82 | 73.36 |
| ResNet-101 | AI | 76.63 | 73.77 | 73.67 | 61.76 | 61.44 | 62.45 | 42.73 | 46.26 | 44.41 |
| | AN | 85.55 | 80.89 | 81.63 | 64.50 | 62.25 | 65.29 | 51.67 | 53.98 | 68.82 |
| | ASL [29] | 88.38 | 82.90 | 84.60 | 69.33 | 66.16 | 68.23 | 54.47 | 56.51 | 69.55 |
| | SPLC [46] | 89.13 | 84.31 | 86.2 | 72.39 | 69.25 | 71.68 | 55.25 | 57.10 | **70.81** |
| | LL [17] | 87.24 | 82.16 | 84.08 | 70.86 | 68.03 | 68.17 | 54.28 | 56.44 | 70.41 |
| | PASL [1] | 88.97 | 83.79 | 85.22 | 70.84 | 67.45 | 69.38 | 54.68 | 56.25 | 69.94 |
| | Selective [1] | 78.72 | 77.25 | 73.07 | 72.85 | 70.17 | 72.61 | 54.03 | 56.05 | 70.42 |
| | MLdecoder [31] | 88.03 | 83.12 | 84.39 | 72.16 | 68.63 | 71.14 | 55.09 | 56.62 | 70.07 |
| | BoostLU [18] | 88.70 | 83.96 | 85.86 | 71.78 | 69.17 | 69.15 | 54.44 | 57.34 | 70.41 |
| | APL (Ours) | **89.94** | **85.17** | **87.14** | **73.52** | **70.33** | **73.38** | **56.28** | **57.55** | 70.67 |
| | APL-EMA (Ours) | 89.94 | 85.17 | 87.14 | 78.49 | 74.49 | 76.61 | 60.02 | 60.94 | 72.70 |

where $\tau_{top}$ and $\tau_{bottom}$ are the top and bottom boundaries of the threshold, respectively. Small $\tau_{c_i}$ means $c_i$ is hard to learn, indicating that fewer false negatives in unknown labels should be considered in the model optimization. As $\tau_{c_i}$ grows, more negative samples are encouraged to be learned.

Instead of using a fixed value in HPL, dynamic thresholds are used to divide unknown labels:

$$U_I^{Neg} = \left\{ c_i | c_i \in U_I, p_{c_i} <= \tau_{c_i} \right\},$$
$$U_I^{Ign} = \left\{ c_i | c_i \in U_I, p_{c_i} > \tau_{c_i} \right\}. \quad (13)$$

The final loss of APL is the same as Eqn. (10). The thresholds are dynamically adjusted at each iteration. As shown in Fig 4.(d), dynamic thresholds are applied.

## 5 Experiments

### 5.1 Experimental Settings

***Datasets.*** Several widely used multi-label benchmarks, VOC-2012, COCO2014, NUSWIDE, and OpenImages, are used to analyze the performance of our proposed setting and method. To generate *block diagonal labels* datasets, we randomly select only one block that has at least a positive label. For fair comparisons, the same random seed is used to create the datasets in all experiments.

We show the block divisions on VOC2012, COCO2014, and NUSWIDE in the Appendix. There are 9 blocks, 11 blocks, and 12 blocks for VOC2012, COCO2014, and NUSWIDE, respectively. (1) VOC2012 contains 5,717 training images with 20 categories and 5,823 images for testing. (2) COCO2014 consists of 82,081 training images with 80 categories and a test set of 40,137 images. (3)

NUSWIDE contains a total of 269,648 images with 81 categories. (4) OpenImages is a large-scale multi-label dataset. According to the category hierarchy, there are 68 blocks which contain 520 categories. 1,211,648 images are used for training and 80,356 images are used for testing.

As for *partial labels*, we randomly discard a fixed portion of labels per image following [13]. We maintain a similar proportion of known labels in *partial labels* as in *block diagonal labels*. The annotation proportions for VOC2012, COCO2014, and NUSWIDE are nearly 11%, 10%, and 16%, respectively. Comparison between *block diagonal labels* and *partial labels* is provided in the Appendix.
***Evaluation Metrics.*** To fairly compare with other methods, we employ the mean Average Precision (mAP), per-category F1-score (CF1), and overall F1-score (OF1) as evaluation metrics.
***Implementation Details.*** We employ the TResNet-Large [30] and ResNet-101 [15] architectures, which are pre-trained on ImageNet [9] dataset, as our backbone networks. Our training configuration modified from [1]. The model is trained for 30 epochs using Adam optimizer [19] with true-weight-decay [26] of 1e-4 and 1-cycle cosine annealing policy [33]. The maximal learning rate is 2e-5 for VOC2012 and 6e-5 for COCO2014 and NUSWIDE, respectively. We scale the learning rate according to the batch size with the formula: $\text{lr}_{scaled} = \frac{\text{lr}}{64} \times$ batchsize. We resize the input images to $448 \times 448$. For data augmentation, we use the Cutout [11] with a factor of 0.5 and the auto-augment [8]. We use 4 GPUs with batch size 128.

For all benchmarks, we use $\gamma_+ = 0$, $\gamma_- = 4$, $\gamma_u = 6$ in Eqn. (10). We use the HPL in the first epoch as a warmup phase. Hyperparameters in Eqn. (12) are as followed: In VOC2012, $\tau_{bottom} =$

**Table 2: DualCoOp [34] with our proposed APL in *block diagonal labels*. The results evaluate the effectiveness on powerful vision-languange CLIP [28].**

| Backbone | Method | VOC2012 | | | COCO2014 | | | NUSWIDE | | |
|---|---|---|---|---|---|---|---|---|---|---|
| | | mAP | CF1 | OF1 | mAP | CF1 | OF1 | mAP | CF1 | OF1 |
| CLIP (ResNet-101) | DualCoOp | 83.16 | 80.66 | 70.52 | 72.99 | 70.24 | 71.32 | 51.83 | 54.15 | 57.69 |
| | DualCoOp+APL(Ours) | 92.63 | 88.21 | 89.53 | 79.53 | 75.53 | 78.33 | 57.38 | 58.69 | 70.20 |
| | | +9.47 | +7.55 | +19.01 | +6.54 | +5.29 | +7.01 | +5.55 | +4.54 | +12.51 |

**Table 3: OpenImages results with TResNet-Large backbone in *block diagonal labels*. EMA indicates the exponential moving average strategy.**

| Method | OpenImages | | |
|---|---|---|---|
| | mAP | CF1 | OF1 |
| AI | 51.28 | 54.05 | 29.94 |
| SPLC [46] | 71.01 | 69.81 | 70.34 |
| Selective [1] | 70.50 | 69.68 | 70.29 |
| APL (Ours) | **72.16** | **70.98** | **70.69** |
| APL-EMA (Ours) | 75.01 | 73.14 | 72.85 |

0.65, $\tau_{top} = 0.85$. In COCO2014, $\tau_{bottom} = 0.25$, $\tau_{top} = 0.90$. In NUSWIDE, $\tau_{bottom} = 0.60$, $\tau_{top} = 0.90$. The length of the memory bank $Z$ is set to 400. Following [1], we apply the exponential moving average strategy (EMA) with a decay of 0.9997 for higher performance. Unless stated otherwise, the experiment results are without EMA.

***Compared Methods.*** For a fair comparison, the training configuration of compared methods is the same as ours. We apply hyperparameter searches to determine the optimal parameters. Only parameters different from the default values of the published paper will be listed.

Two naive baselines mentioned in Sec 4.1 of the main paper, all ignore (**AI**) and all negative (**AN**), use BCE loss. **ASL** applies the asymmetric loss for **AN**, where $\gamma_+ = 0$, $\gamma_- = 4$. **MLdecoder** uses full-decoding ML-decoder structure as the classification head based on **ASL**. $\tau$ in **SPLC** is 0.65. Three versions of **LL** are conducted and the best performance of them is listed in Table 1 of the main paper. In **LL-R**, $\Delta_{rel}$ is 0.2, 0.3, and 0.1 for VOC2012, COCO2014, and NUSWIDE. In **LL-C$_t$**, $\Delta_{rel}$ is always 0.1. In **LL-C$_p$**, $\Delta_{rel}$ is 0.3, 0.5, and 0.3 for VOC2012, COCO2014, and NUSWIDE. **BoostLU** is based on **LL-C$_p$**. **PASL** applies the partial asymmetric loss (Eqn.8 of the main paper), where $\gamma_+ = 0$, $\gamma_- = 4$, $\gamma_u = 6$. **Selective** is based on **PASL**: In VOC2012, $\eta = 0.6, K = 100$. In COCO2014, $\eta = 0.6, K = 50$. In NUSWIDE, $\eta = 0.3, K = 20$. **Selective** uses **AI** to estimate the class distribution in the first stage.

## 5.2 Comparison with State-of-the-art

Table 1 shows the *block diagonal labels* results on VOC2012, COCO-2014, and NUSWIDE with TResNet-Large and ResNet-101 backbones. We compare our proposed APL with state-of-the-art methods. ASL and MLdecoder-ASL treat unknown labels as negatives in the experiments. In the first stage of Selective, the label prior is estimated. Although Selective with TResNet-Large reaches the best performance on COCO2014, the results in other experiments are unstable. By adopting the TResNet-Large backbone, APL surpasses other methods by 1.09% and 0.82% mAP on VOC2012 and NUSWIDE. APL also achieves comparable performance on COCO2014. APL

**Table 4: Comparison with state-of-the-art *single positive labels* methods. FL denotes *full labels*, SPL denotes *single positive labels*, and BDL denotes *block diagonal labels*. # Positive denotes the number of positively annotated labels and # Annotated denotes the number of all annotated labels. † denotes the exponential moving average strategy. * denotes methods based on vision-language pretrained CLIP.**

| Setting | Method | COCO2014 | | |
|---|---|---|---|---|
| | | mAP | # Positive | # Annotated |
| FL | PASL | 83.2 | 241035(100%) | 6566480(100%) |
| SPL | VLPL* [41] | 71.5 | 82081(34.05%) | 82081(1.25%) |
| | MIME [24] | 72.9 | | |
| | DualCoOP* [34] | 73.5 | | |
| | HSPNet*† [36] | 75.7 | | |
| | SCPNet*† [12] | 76.4 | | |
| | Our APL† | 78.3 | | |
| BDL | Our APL† | **81.9** | 107881(44.76%) | 611198(9.31%) |

with ResNet-101 outperforms all existing approaches. The experiments with different backbones validate the robustness of our method. Moreover, EMA averages model weights [35] to further improve the performance of APL. Specifically, APL-EMA reaches 81.92% and 63.26% mAP on COCO2014 and NUSWIDE, respectively.

The results on OpenImages, which is a large-scale multi-label dataset, are shown in Table 3. We compare our APL with the two methods that perform well in Table 1. Our APL outperforms other methods by 1.15% mAP. In addition, our APL with EMA achieves 75.01% mAP.

In Table 2, we conduct further evaluation based on powerful vision-language models. DualCoOp [34] adapts the knowledge acquired in CLIP [28] to multi-label classification with *missing labels*. Although DualCoOp employs prompts to provide positive and negative contexts, this approach overlooks unknown labels during training which may leads to a sub-optimal issue [1]. Notably, DualCoOp enhanced with our APL surpasses the standard DualCoOp by a considerable margin of 6.54% mAP on COCO2014.

## 5.3 Comparison with Single Positive Labels

(1) SPL can be considered an extreme case where each block in BDL contains only one class, converting a multi-label dataset into a single-label dataset. (2) Such extreme setting potentially harm performance in practical applications due to insufficient supervision. Table 4 indicates a notable gap between SPL and FL. However, our APL method in BDL, with 10.71% more positive labels, achieves performance close to FL. (3) We further compare state-of-the-art SPL methods. Despite some methods leveraging vision-language pertained CLIP, our APL method still achieves better results.

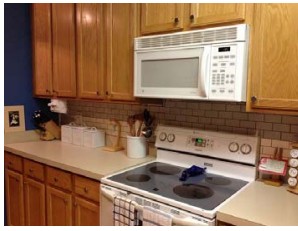 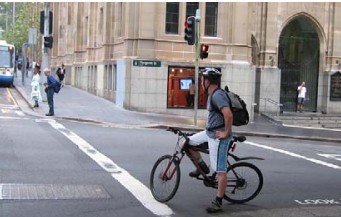 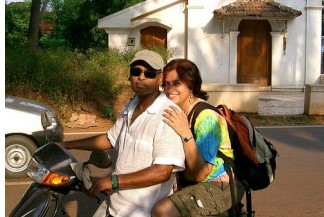 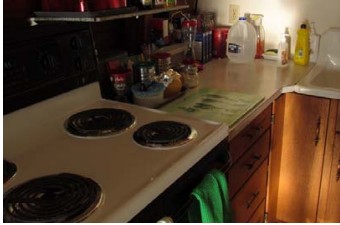

Labeled: microwave, oven
HPL: microwave, oven
APL: microwave, oven, knife, spoon

GT: microwave, oven, knife, spoon,
cup

Labeled: bicycle, bus
HPL: bicycle, bus, person, traffic light
APL: bicycle, bus, person, traffic light,
backpack, handbag

GT: bicycle, bus, person, traffic light,
backpack, handbag, tv

Labeled: car, motorcycle
HPL: car, motorcycle, person
APL: car, motorcycle, person, backpack

GT: car, motorcycle, person, backpack

Labeled: oven, sink
HPL: oven, sink, bottle, bowl, cup, knife
APL: oven, sink, bottle, bowl, spoon,
cup, knife, toaster

GT: oven, sink, bottle, bowl, spoon

**Figure 5: Qualitative results in *block diagonal labels*. GT indicates the ground truth labels for a training image. Labeled indicates the categories labeled in *block diagonal labels*. HPL utilizes hard thresholds while APL employs dynamic thresholds.**

**Table 5: Ablation studies of our APL in *block diagonal labels* and *partial labels* on different benchmarks. The evaluation metric is mAP.**

| Setting | Method | VOC2012 | COCO2014 | NUSWIDE |
|---------|--------|---------|----------|---------|
| *block diagonal labels* | PASL | 90.47 | 76.39 | 58.87 |
| | HPL | 91.22 | 77.79 | 59.37 |
| | APL | 91.56 | 78.25 | 59.69 |
| *partial labels* | PASL | 78.88 | 67.45 | 51.43 |
| | HPL | 84.91 | 71.36 | 54.41 |
| | APL | 86.61 | 71.79 | 54.87 |

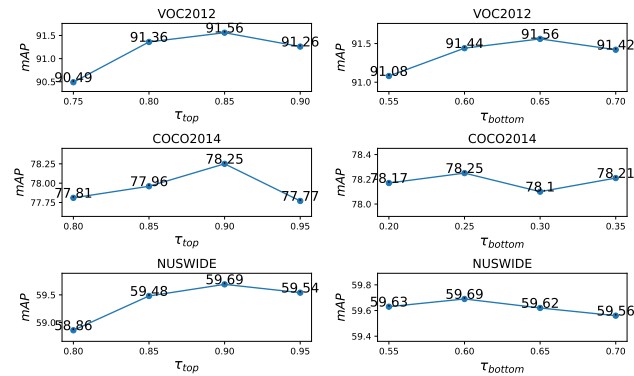

**Figure 6: Ablation studies of $\tau_{top}$ and $\tau_{bottom}$ on different benchmarks. The evaluation metric is mAP.**

## 5.4 Ablation Study

***The Effectiveness of Individual Components.*** In Table 5, we validate the effectiveness of each component in *block diagonal labels* and *partial labels*. We generate the *partial labels* datasets with a similar annotation proportion as the *block diagonal labels* ones, as detailed in the Appendix.

Our proposed HPL and APL apply PASL loss to deal with the positive-negative imbalance issue. HPL sets a constant threshold to find true negatives and reduces the number of false negatives to mitigate performance. However, the thresholds of each category should be dynamically adjusted. Our APL flexibly adjusts thresholds for different classes, which improves the performance in all ablation experiments.

***Hyper-parameters Analysis.*** We investigate the effectiveness of hyper-parameters. As shown in Fig 6, the first col shows the impact of $\tau_{top}$ when $\tau_{bottom}$ is fixed and the second col shows the impact of $\tau_{bottom}$ when $\tau_{top}$ is fixed. Despite variations in values across the three datasets, performance remains consistently near the optimal values. In practice, we can initially set $\tau_{top}$ at 0.9 and conduct a grid search for $\tau_{bottom}$. Once a satisfactory $\tau_{bottom}$ is obtained, we proceed with a grid search for $\tau_{top}$. Fig 6 illustrates the robustness of the hyper-parameters, highlighting the efficacy and convenience of identifying optimal hyper-parameters.

***Qualitative Results.*** We show the qualitative results of HPL and APL in Fig 5. GT indicates the ground truth labels for a training image. Labeled indicates the categories labeled in *block diagonal labels*.

The pseudo labels are marked in blue or red. Blue indicates a correct prediction and red indicates an incorrect prediction. Four cases show that our APL predicts correct labels better than HPL, which demonstrates the superiority of the dynamic threshold. However, in the last case, some errors still occur in the prediction process. The complex scene contains too many small objects, thus confusing the two approaches to predict the pseudo labels.

## 6 Conclusion

In this paper, we demonstrated the main challenge of the multi-label dataset construction from two aspects i.e. learning and annotating. We introduced a new setting of multi-label classification with *block diagonal labels*, which was considered from both sides to reduce the annotation workload. We compared it with the exiting *missing labels* setting to prove the effectiveness of our setting. Additionally, an adaptive pseudo-labeling method was proposed, which is a simple yet effective method without any label prior or complex architectures. Our APL achieves state-of-the-art performance on different benchmarks. For further studies in our *block diagonal labels*, we plan to utilize the prior of coarse label present in each image to improve the prediction accuracy of unknown labels.

# Acknowledgments

This work was supported by "Pioneer" and "Leading Goose" R&D Program of Zhejiang (No. 2023C01038), National Natural Science Foundation of China (Nos. 61925107, 62271281), Zhejiang Provincial Natural Science Foundation of China under Grant (No. LDT23F01013F01).

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

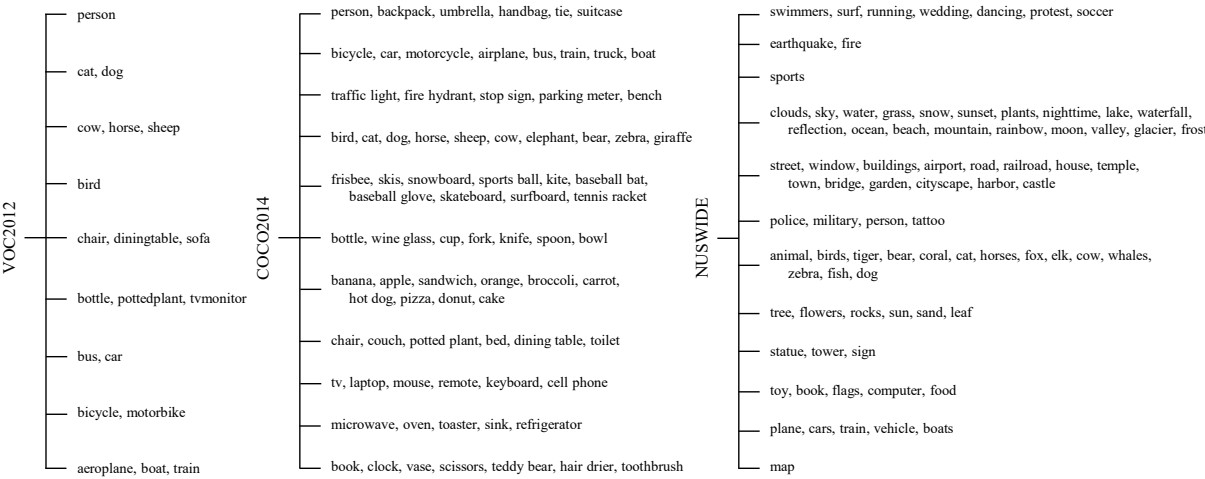

**Figure 7: The categories partitions of *BDL*. The partitions are modified according to the category hierarchies of their published papers. VOC2012 and NUSWIDE have 4 and 6 super-categories, respectively. In order to evaluate the effectiveness of *BDL*, we increase the number of blocks to obtain more unknown labels, so that VOC2012 gets 9 blocks and NUSWIDE gets 12 blocks. As for COCO2014, we use the original partition.**

**Table 6: Comparison of the proportion of annotated labels between *block diagonal labels* and *partial labels*. $\#P_{Pos+Neg}$ indicates the proportion of all annotated labels. $\#P_{Pos}$ indicates the proportion of positive labels.**

| | partial labels | | block diagonal labels | |
|---|---|---|---|---|
| | $\#P_{Pos+Neg}$ | $\#P_{Pos}$ | $\#P_{Pos+Neg}$ | $\#P_{Pos}$ |
| VOC2012 | 10.98% | 0.80% | 10.59% | 5.25% |
| COCO2014 | 9.99% | 0.37% | 9.31% | 1.64% |
| NUSWIDE | 15.99% | 0.48% | 15.60% | 2.01% |

## A  Block Divisions and Details on Benchmarks

Fig 7 illustrates the block divisions on VOC2012, COCO2014, and NUSWIDE. There are 9 blocks, 11 blocks, and 12 blocks for VOC2012, COCO2014, and NUSWIDE, respectively. VOC2012 contains 5,717 training images with 20 categories and 5,823 images for testing. COCO2014 consists of 82,081 training images with 80 categories and a test set of 40,137 images. NUSWIDE contains a total of 269,648 images with 81 categories.

OpenImages is a large-scale multi-label dataset. According to the category hierarchy, there are 68 blocks which contain 520 categories. 1,211,648 images are used for training and 80,356 images are used for testing.

## B  Comparison between Block Diagonal Labels and Partial Labels

To evaluate the superiority of *block diagonal labels*, we compare the experiments of *block diagonal labels* with those of *partial labels*. As

illustrated in Table 6, the number of annotated labels is almost the same between *block diagonal labels* and *partial labels*. We generate the *block diagonal labels* datasets by randomly selecting a block of labels. For comparison between these two settings, we generate the *partial labels* datasets with a similar annotation proportion as the *block diagonal labels* ones. The positive proportions $\#P_{Pos}$ of them differ greatly. For example, $\#P_{Pos}$ of *block diagonal labels* is greater than that of *partial labels* by 4.45% on VOC2012. Randomly annotating a fixed percentage of labels in *partial labels*, which may generate images with all negative labels, will result in a very small number of positives.

Fig 8 shows results of the state-of-the-art methods in *block diagonal labels* and *partial labels*. The results in our *block diagonal labels* significantly surpass the results in *partial labels* on all benchmarks, which proves the superiority of our setting. Especially for SPLC, the performance gaps between these two settings are remarkable. Evident 19.13%, 11.94%, and 8.46% mAP gains are achieved on VOC2012, COCO2014, and NUSWIDE, respectively. The recent approaches treat most unknown labels as negatives. This assumption is mostly correct due to the large number of true negatives. The effect of annotated negative labels is reduced, while the number of positive labels has a significant impact on the final performance. Therefore, the *block diagonal labels* setting not only reduces the annotation workload from both learning and annotating aspects but also increases the number of positives, leading to a notable enhancement in performance.

In addition, on all benchmarks, we find the performance gaps of our APL are relatively small compared to other methods. Our APL outperforms other methods by a large margin in all experiments. Such results prove the effectiveness and robustness of our APL.

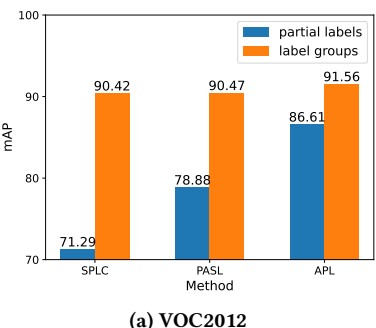
(a) VOC2012

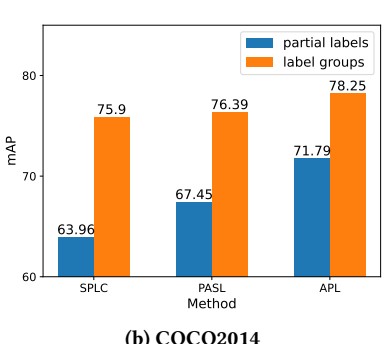
(b) COCO2014

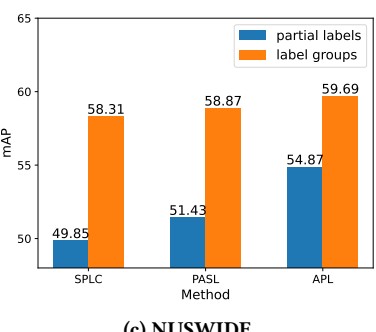
(c) NUSWIDE

Figure 8: Experimental results of the state-of-the-art methods in *block diagonal labels* and *partial labels*. The experiments in *partial labels* are marked in blue. The experiments in *block diagonal labels* are marked in orange.