# OpenReview forum: "Multi-Label Learning with Block Diagonal Labels"
_acmmm.org/ACMMM/2024/Conference — MM2024 Poster_

### Official Review · Reviewer_aGKK · 2024-05-14

[review text omitted: it was posted to a different submission]

---

### Official Review · Reviewer_gZ4t · 2024-05-23

**Rating:** 5
**Confidence:** 3

**Summary:**

This paper proposed a new multi-label learning method that tackle the high annotation worekload problem with block diagonal labels.

**Strengths:**

- The paper is clearly written and easy to follow.
- The idea is interesting and the solution is technical.

**Limitations:**

- A suggestion. I think that the introduction of both learning and annotating in abstract is not easy to be understood for readers. Maybe, you can say different annotators have different knoelwdge background so that they are good at annotating different types of labels. Or, an annotator to annotate examples with few number of categories would achieves better performance than that with large number of labels.

- The experiments. The number of classes in each datasets is not large. You should conduct experiments on MLL datasets with extreme labels.

**Suitability:**

3

---

### Official Review · Reviewer_mMkW · 2024-05-23

**Rating:** 4
**Confidence:** 3

**Summary:**

This paper addresses the challenge of annotating large-scale multi-label datasets by proposing a new setting called block diagonal labels. The method aims to reduce the annotation workload by dividing categories into subsets based on semantics and relevance, allowing annotators to focus on specific and highly relevant subsets. The authors also introduce an Adaptive Pseudo-Labeling (APL) method to predict unknown labels via dynamically adjusting thresholds. Extensive experiments on benchmarks like VOC2012, COCO2014, NUSWIDE, and OpenImages demonstrate the effectiveness of the proposed method, achieving significant improvements in mean Average Precision (mAP), per-category F1-score (CF1), and overall F1-score (OF1), with a reduced annotation effort.

**Strengths:**

The introduction of block diagonal labels to reduce annotation workload is an innovative approach that addresses a practical problem in multi-label classification.

The proposed setting significantly reduces the number of annotations required, making it feasible to create large-scale multi-label datasets.

The introduction of APL, which dynamically adjusts label prediction thresholds, enhances model performance, as well as improving the applicability of the block diagonal labels in real-world practice.

The paper provides a detailed explanation of the methodology, ensuring transparency and reproducibility of the experiments.

**Limitations:**

The effectiveness of the block diagonal labels setting heavily relies on how well the categories are partitioned, which may be challenging in real-world scenarios where label distribution is not as easily controlled or partitioned.

Although the paper shows that the proposed APL works well with the block diagonal labels, it doesn't consider the difference of importance of each class. In practice, certain classes are much more important to be labelled than others, for example, missing labelling a bin might not matter much but missing labelling a pedestrian might cause catastrophic result.

**Suitability:**

3

---

### Official Review · Reviewer_nrgQ · 2024-05-26

**Rating:** 2
**Confidence:** 4

**Summary:**

This paper proposes a new setting to reduce the annotation cost.

**Strengths:**

1.The paper is well-written and easy to follow.

2.The analysis of the methods section is very good, and the descriptions of various strategies are very detailed.

**Limitations:**

1.Insufficient research on related work. Based on my knowledge, the setting of biased markers is not limited to the one mentioned in this work. Analyzing the difference between literature [1] and your setting. It is not inevitable that some samples lack positive labels.

2.The discussion and analysis of single positive multi-labels learning are also insufficient, and the references are too outdated, seemingly all before 2022. In fact, many relatively new jobs have already achieved good results [2,3], which is inconsistent with the shortcomings of single positive multiple labeling discussed by the author in Chapter 3.

3.The experimental part is unfair, and the single positive multi labeling method used for comparison is too outdated. The author should compare it with some new methods. In addition to references [2,3], as far as I know, there are still many methods based on vision-language pretained (VLP) models. By utilizing the transfer ability of VLP models, it can effectively compensate for the shortcomings brought by single positive labels. The author's experiments on VLP models are also not comprehensive.

[1] Partial Multi-Label Learning with Meta Disambiguation. KDD 2021.

[2] Revisiting Pseudo-Label for Single-Positive Multi-Label Learning. ICML 2023.

[3] Hierarchical Prompt Learning Using CLIP for Multi-label Classification with Single Positive Labels. MM 2023.

**Suitability:**

2

---

### Meta-Review · Area_Chair_BjQp · 2024-06-30

**Recommendation:** Accept (Poster)
**Confidence:** 5

**Metareview:**

This paper introduces a method to reduce annotation workload in large-scale multi-label datasets. Concerns were raised regarding experimental fairness and other issues. All reviewers agree on the merits of the work. The ACs support this assessment and recommend accepting the paper. Please incorporate reviewers' suggestions into the camera-ready version to enhance the paper's clarity and include new results from the rebuttal.